# ExplicitLM: Decoupling Knowledge from Parameters via Explicit Memory Banks

## Abstract

Large language models (LLMs) universally suffer from knowledge staleness and lack of interpretability due to their implicit knowledge storage paradigm, where information is distributed across network parameters in an entangled, non-addressable manner. This fundamental limitation prevents targeted knowledge updates, verification of stored information, and understanding of model reasoning processes. We propose ExplicitLM, a novel architecture that fundamentally reimagines knowledge storage in language models through an explicit, interpretable memory bank system. Our key innovation introduces a million-scale external memory bank where each entry stores human-readable knowledge as token sequences, enabling direct inspection and modification of the model's knowledge base. To efficiently access this massive repository, we design a differentiable two-stage retrieval mechanism that enables end-to-end training while maintaining discrete knowledge selection, combining efficient coarse-grained filtering with product key decomposition (reducing computational complexity from $\mathcal{O}(N \cdot |I|)$ to $\mathcal{O}(\sqrt{N} \cdot |I|)$) and fine-grained similarity matching through Gumbel-Softmax. Drawing inspiration from dual-system cognitive theory, we partition knowledge into frozen explicit facts (20%) and learnable implicit patterns (80%), maintained through an Exponential Moving Average update strategy that ensures training stability. Extensive experiments demonstrate that ExplicitLM achieves up to 43.67% improvement in knowledge-intensive tasks compared to standard Transformers, with particularly pronounced gains in low-data regimes ($3.62\times$ improvement with 10k samples). Our analysis reveals strong correlations between memory retrieval success and task performance, with correctly predicted samples achieving 49% higher memory hit rates. Unlike traditional RAG systems with frozen retrieval components, our jointly optimized architecture demonstrates that interpretable, updatable language models can maintain competitive performance while providing unprecedented transparency into their knowledge utilization.

## 1 Introduction

Contemporary large language models (LLMs) universally suffer from knowledge staleness, with internally stored knowledge frozen at training completion Cheng et al. (2024); Singh et al. (2025). This temporal limitation creates a widening gap between static model knowledge and dynamic real-world information. Consider the U.S. presidency: Joe Biden served until January 2025, when Donald Trump assumed office. Models trained before this transition perpetually provide outdated answers, unable to reflect real-time changes. Post-training, this knowledge ossification accumulates across countless facts—political leadership, scientific discoveries, economic indicators, and technological standards—severely undermining model reliability in practical applications Mousavi et al. (2024). Knowledge updating thus emerges as critical: models require mechanisms to incorporate temporal factual changes to maintain utility and trustworthiness in real-world deployments.

Current approaches to acquiring or updating external knowledge primarily rely on two paradigms: real-time querying through Model Context Protocol (MCP) toolsHou et al. (2025), or knowledge augmentation via Retrieval-Augmented Generation (RAG) techniquesLewis et al. (2020).However, MCP-based methods exhibit several critical limitations. First, real-time querying introduces substantial inference latency, degrading user experienceSingh et al. (2025). Second, dependency on

external APIs compromises system robustnessLi et al. (2025).RAG techniques, though partially mitigating knowledge updating challenges, face persistent obstacles: the relevance between retrieved documents and queries remains difficult to ensure, the inherent misalignment between retrieval and generation objectives yields suboptimal performance, and the maintenance and updating of external knowledge bases incurs substantial engineering overheadSalemi & Zamani (2024). These limitations collectively motivate the need for more efficient and integrated approaches to knowledge acquisition and updating in language models.

The fundamental barrier to direct manipulation of model-internal knowledge stems from the implicit knowledge storage paradigm in current language models. Research demonstrates that LLM knowledge is predominantly distributed across Feed-Forward Network (FFN) layers of the Transformer architectureGeva et al. (2021); Meng et al. (2022); Dai et al. (2022). Unlike traditional databases with discrete, addressable locations, each piece of LLM knowledge emerges from collective parameter interactions across all FFN layers, creating highly entangled representations that cannot be independently isolated or modified. This transforms knowledge update into a formidable challenge: modifying a single fact theoretically requires recalibrating weights throughout the entire network—a practically infeasible task risking catastrophic interference with other stored knowledge. This "black-box" nature prevents both verification of acquired knowledge and targeted correction of problematic content. During pre-training on massive corpora, models inevitably absorb misinformation, outdated content, or harmful materialPerełkiewicz & Poświata (2024), yet inability to precisely locate and excise such knowledge allows errors to persist and propagate through outputs, fundamentally undermining reliability and trustworthiness.

More critically, implicit knowledge storage fundamentally impedes interpretability. When generating predictions, researchers cannot trace specific knowledge foundations underlying model reasoning. We cannot determine which facts inform reasoning nor verify reasoning step correctness. This opacity constrains understanding of model behavior and poses fundamental challenges to building trustworthy, interpretable AI systems. In high-reliability domains like medical diagnosisEnnab & Mcheick (2024) and legal consultationLatif (2025), this interpretability lack becomes a primary deployment barrier.

Motivated by these observations, we propose a novel language model architecture incorporating an explicit memory bank. The core innovation lies in transforming traditional implicit knowledge storage into an explicit, interpretable knowledge management system. By introducing accessible Memory Bank layers at each model layer, we enable dynamic retrieval and utilization of external knowledge while, more importantly, achieving transparent knowledge management. Our main contributions are summarized as follows:

- We propose an explicit knowledge storage architecture based on Memory Banks, enabling each knowledge entry in the model's repository to be decoded into human-readable text format, fundamentally addressing the interpretability limitations of traditional models.

- We design a differentiable two-stage retrieval mechanism that combines discrete knowledge selection with continuous gradient flow, enabling end-to-end training of the memory-augmented architecture while maintaining retrievable interpretability and low computational cost.

- We propose ExplicitLM, a novel architecture that enables explicit retrieval and interpretation of model knowledge while achieving superior answer accuracy compared to standard Transformer baselines.

## 2 RELATED WORK

### 2.1 LLM ARCHITECTURE DEVELOPMENT

The evolution of large language model architectures began with BERT Devlin et al. (2019), which introduced bidirectional pre-training through masked language modeling, while GPT-2 Radford et al. demonstrated the power of scaling autoregressive transformers. T5 Raffel et al. (2020) unified various NLP tasks into a text-to-text framework, and GPT-3 Brown et al. (2020) showed emergent few-shot learning capabilities at 175B parameters. Subsequent developments include PaLM Chowdhery et al. (2023) scaling to 540B parameters with improved training efficiency, LLaMA Touvron et al.

(2023) achieving strong performance with smaller models through careful data curation, and GPT-4 Achiam et al. (2023) advancing multimodal capabilities. Recent architectural innovations have explored alternatives to standard transformers: RWKV Peng et al. (2023) combines RNN efficiency with transformer-level performance through linear attention mechanisms, Mamba Gu & Dao (2023) leverages selective state space models for efficient long-context modeling with linear complexity, while Mixtral Jiang et al. (2024) employs sparse mixture-of-experts for improved parameter efficiency.

## 2.2 KNOWLEDGE EDITING AND UPDATING

Knowledge editing in large language models to eliminate errors remains an emerging research area, with existing approaches divided into parameter-efficient and parameter-augmented methods. Parameter-efficient approaches focus on updating knowledge without additional parameters: Li et al. (2023) introduces KAFT (Knowledge-Augmented Fine-Tuning), a data augmentation strategy incorporating diverse contexts (relevant, irrelevant, and counterfactual) for fine-tuning to reduce knowledge errors, while Onoe et al. (2023) constructs datasets to evaluate whether different methods can successfully inject specific facts and enable reasoning based on them. Parameter-augmented methods introduce additional components: Dong et al. (2022) employs CaliNet, training key-value calibration memory slots with similar architecture to FFN but smaller intermediate dimensions; Wang et al. (2024) embeds memory pools containing compressed knowledge tokens at each layer with update functions, though lacking interpretability; Mitchell et al. (2022) prepends a knowledge classifier to existing models, routing queries to either an explicitly stored and updatable database with a specialized model or the standard LLM, achieving explicit storage but sacrificing end-to-end neural architecture coherence.

## 3 MEMORY BANK

### 3.1 MEMORY THEORY

Drawing from dual-system cognitive theory Gowda et al. (2025), we partition language model knowledge into two distinct yet complementary phases analogous to human procedural-declarative memory dichotomy.

**Implicit Knowledge**: This encompasses linguistic grammar rules, syntactic structures, and semantic associations that resist explicit formalization. Examples include nuanced aspects of human expression patterns and implicit connections between complex concepts that emerge from cultural and contextual understanding. Such knowledge exhibits high abstraction and ambiguity, necessitating statistical learning from large-scale data.

**Explicit Knowledge**: This comprises factual knowledge, entity relationships, and time-sensitive information amenable to explicit representation. Examples include "The President of the United States is Trump" (not Biden) and "The Eiffel Tower stands 324 meters tall." Such knowledge possesses clear truth conditions and update requirements, making it suitable for storage in editable memory banks.

This dual-system design offers distinct advantages: implicit knowledge, acquired through deep learning, ensures robust language understanding and generation capabilities; explicit knowledge, through structured storage, enables interpretability and updatability. The synergistic integration of both systems enables models to maintain powerful linguistic capabilities while flexibly managing and updating factual knowledge.

### 3.2 STORAGE ARCHITECTURE

Let $\mathcal{M} \subseteq \mathbb{Z}^{1 \times L}, |M| = N$ denote our Memory Bank tensor, where $N = 10^6$ represents the knowledge capacity and $L = 16$ denotes the maximum sequence length. Each entry $\mathbf{m}_i \in \mathcal{M}$ stores a discrete knowledge unit as token indices, with elements $m_{ij} \in \mathcal{V}$, where $\mathcal{V}$ is codebook.

We employ a tokenizer-based bidirectional mapping scheme. The encoding function Tokenize : $\mathcal{S} \to \mathbb{Z}^{1 \times L}$ converts knowledge strings $s \in \mathcal{S}$ to token indices for storage: $\mathbf{m}_i = \text{Tokenize}(s_i) = [t_1^{(i)}, t_2^{(i)}, ..., t_L^{(i)}]$ where $t_j^{(i)} \in \mathcal{V}$. During retrieval, the embedding function Embed : $\mathbb{Z}^{1 \times L} \to$

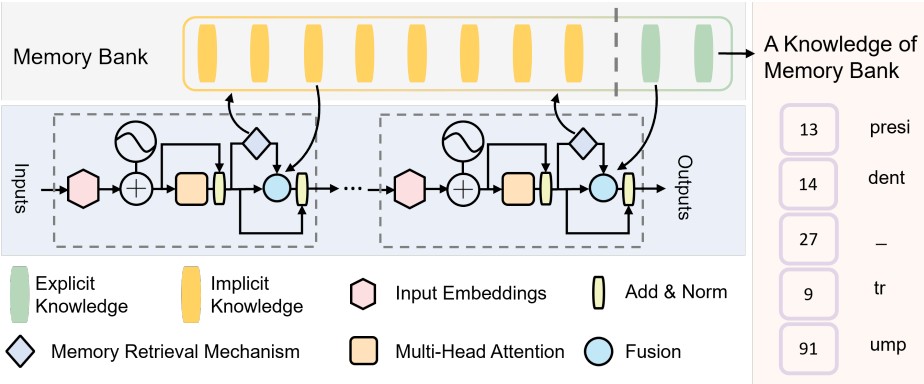

Figure 1: Overall architecture of ExplicitLM. The blue region shows the multi-layer transformer blocks. The gray region represents the shared Memory Bank accessed by all layers, where each layer can retrieve knowledge via the Memory Retrieval Mechanism (Section 3.4) from Explicit Knowledge (green) or Implicit Knowledge (yellow) partitions. The orange region shows a sample knowledge entry from the Memory Bank—a sequence of token indices of length $L$ directly convertible to human-readable text.

$\mathbb{R}^{d \times L}$ transforms stored indices back to semantic representations: $\mathbf{E}_i = \text{Embed}(\mathbf{m}_i) = [\mathbf{e}_{t_1^{(i)}}, \mathbf{e}_{t_2^{(i)}}, ..., \mathbf{e}_{t_L^{(i)}}]$, where $\mathbf{e}_{t_j^{(i)}} \in \mathbb{R}^d$.

### 3.3 KNOWLEDGE ALLOCATION STRATEGY

Given the memory constraint $|\mathcal{M}| = N$, our approach maintains a fixed-capacity knowledge repository throughout the model's lifecycle. This design choice ensures predictable memory consumption and eliminates the computational overhead associated with dynamic memory allocation. To effectively utilize this fixed capacity while preserving essential linguistic knowledge, we introduce a partitioning scheme that divides the memory bank into two disjoint subsets: $\mathcal{M} = \mathcal{M}_f \cup \mathcal{M}_u$ where $\mathcal{M}_f \cap \mathcal{M}_u = \emptyset$. The partition is controlled by a freeze rate parameter $\rho \in [0, 1]$, which determines the proportion of memory allocated to each subset.

The frozen knowledge subset $\mathcal{M}_f$ with cardinality $|\mathcal{M}_f| = \rho N$ (we empirically set $\rho = 0.2$ as default) is designated for storing explicit knowledge that can be precisely formulated and verified. During initialization, this subset is populated with curated factual information such as entity relationships, geographical facts, and time-sensitive data that require accurate representation. The explicit nature of this knowledge allows for direct injection of verified information into the memory bank, ensuring factual accuracy from the outset. These entries remain immutable during training to preserve the integrity of the pre-verified knowledge base. Conversely, the updatable knowledge subset $\mathcal{M}_u$ with cardinality $|\mathcal{M}_u| = (1 - \rho)N$ is allocated for implicit knowledge that the model must discover through training. This subset captures linguistic regularities, syntactic patterns, and semantic associations that emerge from statistical learning over large-scale text corpora. The model autonomously determines which grammatical structures and language patterns warrant storage in this dynamic portion of the memory bank. The in-place substitution mechanism maintains the invariant $|\mathcal{M}^{(t)}| = N$ for all time steps $t$, as updates neither insert new entries nor delete existing ones, thereby preserving constant memory footprint and eliminating the complexity associated with dynamic memory management operations.

To address the gradient discontinuity issue that arises from direct overwriting of knowledge entries in $\mathcal{M}_u$, we adopt the Exponential Moving Average (EMA) technique from Vector Quantized Variational Autoencoders (VQ-VAE) Van Den Oord et al. (2017), originally developed for codebook updates. Rather than performing abrupt replacements, the EMA mechanism enables progressive updates that maintain training stability. Specifically, for each knowledge entry $\mathbf{m}_i \in \mathcal{M}_u$, we maintain dynamic statistics that allow smooth transitions between old and new knowledge representations. The update rule assigns higher weights to newer information while preserving continuity with existing knowledge, enabling the memory bank to adapt to evolving encoder outputs without

introducing disruptive oscillations. This approach effectively circumvents the non-differentiability inherent in discrete quantization operations, while simultaneously improving both the utilization rate of knowledge entries and the overall reconstruction quality of the stored information.

## 3.4 MEMORY RETRIEVAL MECHANISM

We propose a hierarchical two-stage retrieval strategy for efficient access to million-scale entries.

Figure 2: ExplicitLM architecture with memory retrieval mechanism. In Stage 1, both query and key vectors are partitioned along the embedding dimension into two components for efficient retrieval. In Stage 2, cosine similarity is computed between the query and candidate knowledge entries, with the highest-scoring entry selected for retrieval.

**Stage 1: Key-value Filtering.** Following Million Experts He (2024), we assign product keys $\mathbf{K} := \{k_i\}_{i=1}^N \subset \mathbb{R}^d$ to knowledge entries, with query network $q$ mapping input $x$ to query $q(x)$. This stage generates a candidate set $I$ by retrieving the most relevant entries based on query-key similarities: $I = \text{top-}I\text{-indices}\left(\{q(x)^\top k \mid k \in \mathbf{K}\}\right)$, where top-$I$-indices denotes the operator that selects the indices of the top-$I$ elements from $\mathbf{K}$, yielding a candidate set with cardinality $|I|$. To address computational complexity at $N \geq 10^6$, we decompose keys using Cartesian products: $\mathbf{K} = \{[c; c'] \mid c \in \mathbf{C}, c' \in \mathbf{C}'\}$ where $\mathbf{C}, \mathbf{C}' \subset \mathbb{R}^{d/2}$ with $|\mathbf{C}| = |\mathbf{C}'| = \sqrt{N}$, reducing complexity from $\mathcal{O}(N \cdot |I|)$ to $\mathcal{O}(\sqrt{N} \cdot |I|)$, where $|I| \ll \sqrt{N}$.

**Stage 2: Similarity Selection.** For candidates $i \in I$, we compute cosine similarities $cs_i = \cos(q(x), k_i)$ and apply Gumbel-Softmax for differentiable selection:

$$p_i = \frac{\exp\left((cs_i + g_i)/\tau\right)}{\sum_{j \in I} \exp\left((cs_j + g_j)/\tau\right)} \tag{1}$$

where $g_i = -\log(-\log(\epsilon_i))$ with $\epsilon_i \sim \text{Uniform}(0,1)$ and temperature $\tau$. The straight-through estimator enables gradient flow: forward pass selects $\mathbf{m}_{\text{selected}} = \mathbf{m}_{\hat{i}}$ where $\hat{i} = \arg\max_i p_i$, while backward pass uses soft weights $\frac{\partial \mathcal{L}}{\partial q(x)} = \sum_{i \in I} p_i \frac{\partial \mathcal{L}}{\partial \mathbf{m}_i}$, maintaining discrete selection while ensuring end-to-end differentiability for retrieved knowledge $\mathbf{m}_{\text{selected}} \in \mathcal{M}$.

Unlike traditional RAG systems that rely on frozen retrieval components, our mechanism enables joint optimization of retrieval and generation through differentiable selection, allowing the model to learn task-specific retrieval patterns during training.

## 3.5 JOINT OPTIMIZATION OBJECTIVE

We design a multi-task learning framework that jointly optimizes three complementary losses to balance language modeling capability with effective memory retrieval.

**Language Modeling Loss.** Following standard practice, we minimize cross-entropy between predicted and ground-truth distributions. For sequence $\mathbf{x} = (x_1, \ldots, x_T)$ with vocabulary $\mathcal{V}$ of size $V$:

$$\mathcal{L}_{\text{CE}} = -\frac{1}{T} \sum_{t=1}^T \log p(x_t | x_{<t}, \mathcal{M}) \tag{2}$$

where $p(x_t|x_{<t}, \mathcal{M})$ denotes the model's predicted probability conditioned on context $x_{<t}$ and retrieved memories from $\mathcal{M}$.

**Memory Relevance Loss.** To ensure semantic alignment between queries and retrieved memories, we maximize weighted cosine similarities. Given query $q(x) \in \mathbb{R}^d$ and retrieved candidates $\{\mathbf{E}_i\}_{i=1}^{|I|}$ with selection weights $\{p_i\}_{i=1}^{|I|}$ from Gumbel-Softmax:

$$\mathcal{L}_{\text{sim}} = -\mathbb{E}_{\mathbf{x} \sim \mathcal{D}} \left[ \sum_{i=1}^{|I|} p_i \cdot \frac{q(\mathbf{x})^T \mathbf{E}_i}{\|q(\mathbf{x})\|_2 \|\mathbf{E}_i\|_2} \right] \tag{3}$$

This loss guides the model toward selecting contextually relevant memories by reinforcing high-similarity retrievals.

**Memory Diversity Loss.** To prevent retrieval collapse into local regions and expand semantic coverage, we minimize pairwise similarities among the candidates. Let $\hat{\mathbf{E}}_i = \mathbf{E}_i / \|\mathbf{E}_i\|_2$ denote normalized embeddings:

$$\mathcal{L}_{\text{div}} = \frac{2}{|I|(|I|-1)} \sum_{i=1}^{|I|} \sum_{j=1, j \neq i}^{|I|} cs(\hat{\mathbf{E}}_i, \hat{\mathbf{E}}_j) \tag{4}$$

This regularization encourages exploration across diverse memory regions, preventing locally optimal retrieval patterns while maintaining relevance through balanced optimization.

The final objective combines all losses: $\mathcal{L}_{\text{total}} = \mathcal{L}_{\text{CE}} + \lambda_{\text{sim}} \mathcal{L}_{\text{sim}} + \lambda_{\text{div}} \mathcal{L}_{\text{div}}$. This joint optimization ensures: (1) accurate next-token prediction through $\mathcal{L}_{\text{CE}}$, (2) semantically coherent retrieval via $\mathcal{L}_{\text{sim}}$, and (3) diverse memory exploration through $\mathcal{L}_{\text{div}}$, yielding an end-to-end trainable knowledge-augmented architecture where memory retrieval and language modeling are deeply integrated.

## 4 EXPERIMENTS

### 4.1 DATASET CONSTRUCTION

We construct a 10M-entry multi-source pretraining corpus with strategic sampling ratios optimized for knowledge diversity: **Wikipedia**: Structured encyclopedic knowledge annotated with entity triplets for explicit knowledge graph extraction. These entries form the exclusive source for Memory Bank initialization $\mathcal{M} \subseteq \mathbb{Z}^{1 \times L}$, selected based on knowledge density metrics and factual reliability scores. **Project Gutenberg**: Literary and historical texts providing formal language patterns and narrative structures spanning multiple centuries. **OpenWebText**: Contemporary web text capturing modern linguistic phenomena and informal discourse patterns.

Each entry maintains a unique identifier for provenance tracking. The Memory Bank entries $\mathbf{m}_i$ are mapped to source UUIDs, enabling systematic knowledge updates and verification. Selection criteria prioritize: (i) token-level information density, (ii) factual accuracy via cross-reference validation, and (iii) domain coverage measured by entity distribution.

### 4.2 EVALUATION TASK DESIGN

We design three complementary tasks to evaluate knowledge utilization from Memory Bank $\mathcal{M}$: **(i) Object Prediction**: Given subject-predicate pairs from knowledge entries $\mathbf{m}_i \in \mathcal{M}$, predict correct object tokens $t_{ji}$ from candidate set. Accuracy measures entity relationship understanding with 5 distractors in $\mathbb{R}^d$ space. **(ii) Relation Reasoning**: Given entity token pairs $(t_{ji}, t_{ki})$ from $\mathbf{m}_i$, infer their semantic relationship. This probes compositional reasoning over stored knowledge structures in $\mathcal{M}$. **(iii) Fact Verification:** Binary classification of statements derived from memory bank domain. Negative samples generated via token substitution at indices $m_{i,j}$ maintain $50:50$ class balance. Data partitioning leverages freeze partition: test samples derive exclusively from frozen entries $\mathcal{M}_f$ where $|\mathcal{M}_f| = \rho N$, while training excludes all tokens from $\mathbf{m}_i \in \mathcal{M}_f$. This strict disjoint constraint between $\mathcal{M}_f$ and training data prevents memorization-based evaluation inflation.

## 4.3 COMPARISON OF DIFFERENT DATA VOLUMES

To systematically evaluate the efficacy of our memory-augmented architecture, we conduct controlled experiments across varying supervised fine-tuning (SFT) data volumes. Both our model and the baseline Transformer undergo identical optimization procedures, with performance assessed on the three tasks defined in Section 4.2. The baseline represents a standard Transformer architecture without memory augmentation, enabling direct attribution of performance gains to our proposed Memory Bank mechanism.

Table 1: Performance comparison between baseline Transformer and our memory-augmented model across different SFT data volumes. Results show accuracy (%) on three knowledge-intensive tasks.

| Data Volume | Model | Object Prediction | Relation Reasoning | Fact Verification |
|---|---|---|---|---|
| 10k | Baseline | 7.86% | 38.27% | 61.71% |
| | Ours | 28.42%↑20.56% | 70.02% ↑31.75% | 66.03%↑4.32% |
| 25k | Baseline | 22.16% | 79.99% | 71.49% |
| | Ours | 63.12%↑40.96% | 87.85% ↑7.86% | 79.79%↑8.33% |
| 50k | Baseline | 30.23% | 83.80% | 83.34% |
| | Ours | 73.90%↑43.67% | 90.41% ↑6.61% | 86.25%↑2.91% |
| 75k | Baseline | 40.64% | 87.66% | 86.40% |
| | Ours | 79.76%↑39.12% | 92.12% ↑4.46% | 88.74%↑2.34% |
| 100k | Baseline | 56.80% | 91.91% | 88.92% |
| | Ours | 80.94%↑24.14% | 92.73% ↑0.82% | 89.75%↑0.83% |

The experimental results reveal pronounced performance advantages in low-data regimes. At 10k training samples, our model achieves 3.62× improvement in Object Prediction and 1.83× improvement in Relation Reasoning compared to the baseline. This substantial gap demonstrates that explicit memory retrieval from $\mathcal{M}$ effectively compensates for limited training exposure, particularly for tasks requiring precise entity-level knowledge recall. The Object Prediction task, which directly queries stored triplets from memory entries $\mathbf{m}_i$, exhibits the most consistent improvements across all data scales (24.14% at 100k samples), validating our retrieval mechanism's effectiveness in accessing specific tokens $t_{ji}$ from the Memory Bank.

## 4.4 MEMORY BANK HIT RATE ANALYSIS

To empirically validate the effectiveness of our Memory Bank retrieval mechanism, we conduct a fine-grained analysis of layer-wise memory access patterns. Using models trained with varying data volumes from Section 1, we examine the correlation between successful memory retrieval and task performance on Relation Reasoning. For each forward pass, we track whether the retrieval mechanism successfully matches relevant entries from $\mathcal{M}$ at each transformer layer, providing insights into how different layers utilize external memory.

The aggregate hit rates reveal a strong correlation between memory access success and prediction accuracy. Models trained on 100k, 50k, 25k, 10k, and 5k samples achieve overall hit rates of 71%, 65%, 66%, 71%, and 71% respectively for correctly answered samples, where a sample is considered to have "hit" if at least one layer successfully retrieves relevant memory. In stark contrast, incorrectly answered samples exhibit substantially lower hit rates of 23%, 21%, 21%, 22%, and 37% respectively. This 3× differential in hit rates between correct and incorrect predictions empirically confirms that successful memory retrieval directly contributes to task performance.

Figure 3 presents the layer-wise decomposition of hit rates, revealing distinct retrieval patterns across the network depth. Both correct and incorrect samples exhibit elevated hit rates at layers L1 and L3, suggesting these layers serve as critical junctures for knowledge integration. The consistency of this pattern across different training data volumes indicates an emergent specialization in the network architecture, where specific layers develop stronger affinity for external memory access.

## 4.5 IMPACT OF FREEZE RATE ON PERFORMANCE

To investigate freeze rate parameter $\rho$ effects on model performance, we conduct systematic experiments varying $\rho$ while maintaining other hyperparameters constant. The freeze rate controls

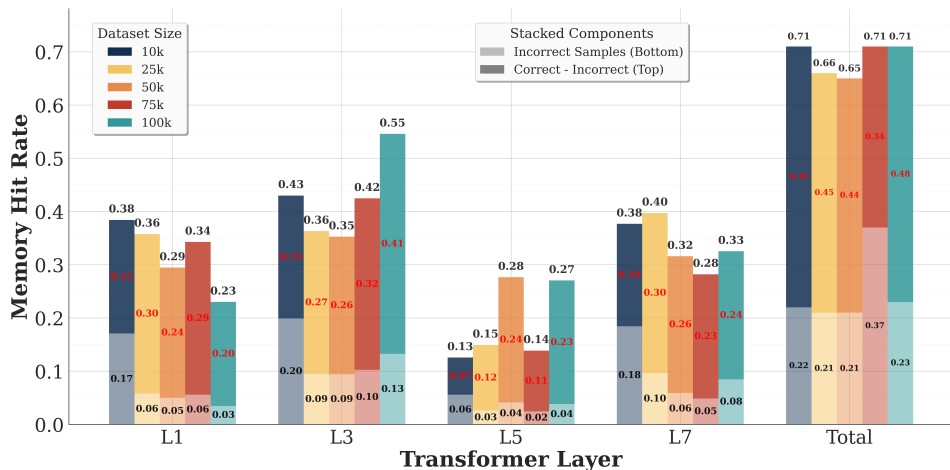

Figure 3: Layer-wise memory hit rates for Relation Reasoning across varying training data volumes. Semi-transparent regions indicate hit rates for correctly predicted samples, while opaque regions show hit rates for incorrect predictions. Red annotations display the hit rate differential between correct and incorrect predictions at each layer.

partition between frozen knowledge $\mathcal{M}_f$ and updatable knowledge $\mathcal{M}_u$, with $|\mathcal{M}_f| = \rho N$ and $|\mathcal{M}_u| = (1 - \rho)N$. We evaluate performance on Relation Reasoning across different training data volumes to understand how explicit-implicit knowledge balance affects learning dynamics.

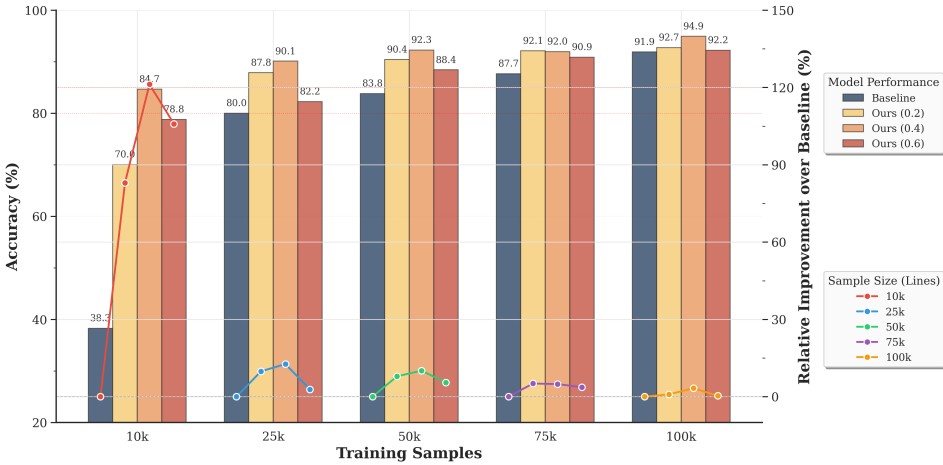

Figure 4: Performance comparison across different freeze rates. The bar chart shows accuracy values under various experimental conditions with different training set sizes. The line plot indicates the relative performance improvement (in percentage) of our method compared to the baseline at different freeze rates for each training set size.

Figure 4 demonstrates that our memory-augmented architecture consistently outperforms the baseline across all freeze rate configurations. Most pronounced improvements emerge in low-data regimes: with 10k training samples, our method achieves minimum 83% improvement regardless of $\rho$, highlighting Memory Bank mechanism robustness to hyperparameter selection. Even at 100k samples where baseline reaches 91.91% accuracy, our approach maintains 0.3%-3.3% improvements, confirming explicit memory benefits persist when parametric learning approaches saturation.

The freeze rate-performance relationship exhibits non-monotonic patterns, with optimal performance at $\rho = 0.4$ across most training set sizes. This peak suggests critical balance between explicit knowledge preservation in $\mathcal{M}_f$ and implicit knowledge adaptation in $\mathcal{M}_u$. Lower freeze

rates ($\rho < 0.4$) potentially compromise core linguistic knowledge stability, allowing excessive updates corrupting fundamental representations. Higher freeze rates ($\rho > 0.4$) restrict model capacity to incorporate task-specific patterns through gradient-based learning, limiting domain-specific adaptation. This trade-off validates our architectural design where frozen entries preserve high-fidelity factual knowledge while updatable entries accommodate evolving linguistic patterns, with optimal partition emerging empirically at approximately 40% frozen knowledge allocation.

### 4.6 IMPACT OF PERFECT RETRIEVAL ON MODEL PERFORMANCE

To quantify the potential performance gains from improved retrieval accuracy, we conduct controlled experiments comparing autonomous retrieval (Retain) against surgical replacement (Replace) of retrieval results. Based on the critical layers identified in Section 4.5, we intervene at layers L1 and L3 by replacing the top-ranked candidate from the 16 retrieved entries with the oracle knowledge entry most relevant to the correct answer. This experimental design isolates the effect of retrieval quality from other architectural components, providing an upper bound on performance improvements achievable through enhanced retrieval mechanisms.

Table 2: Accuracy comparison between autonomous retrieval (Retain) and surgical replacement of retrieval results at specific layers (Replace) to evaluate the impact of perfect retrieval on model performance.

| Data Volume | Model | Object Prediction | Relation Reasoning | Fact Verification |
|---|---|---|---|---|
| 50k | Retain | 70.87% | 89.87% | 85.12% |
| | Replace | 74.49% ↑3.62% | 92.12% ↑2.25% | 87.24% ↑2.12% |
| 75k | Retain | 77.12% | 90.25% | 88.00% |
| | Replace | 79.85% ↑2.73% | 91.87% ↑1.62% | 90.25% ↑2.25% |
| 100k | Retain | 79.12% | 90.50% | 90.37% |
| | Replace | 81.00% ↑1.88% | 92.25% ↑1.75% | 91.12% ↑0.75% |

Table 2 demonstrates consistent improvements across all tasks when perfect retrieval is guaranteed, with an average accuracy gain of 2.11 percentage points. The Object Prediction task exhibits the largest improvements (3.62% at 50k samples), consistent with its direct dependence on retrieving specific factual entries from $\mathcal{M}$. This task directly queries token sequences $\mathbf{m}_i$ for entity relationships, making it most sensitive to retrieval precision. Relation Reasoning shows moderate gains (2.25% at 50k, 1.75% at 100k), suggesting that compositional reasoning benefits from accurate knowledge retrieval but also relies on learned transformations within the network. The diminishing returns observed at larger training volumes (100k samples) indicate that models with more extensive training develop compensatory mechanisms for imperfect retrieval. The average improvement decreases from 2.66% at 50k samples to 1.46% at 100k samples, suggesting that larger training sets enable the model to learn robust representations that partially mitigate retrieval errors.

## 5 CONCLUSION

We presented ExplicitLM, a novel language model architecture that fundamentally transforms knowledge storage from implicit distributed representations to an explicit, interpretable Memory Bank system. Our approach addresses critical LLM limitations—knowledge staleness, lack of interpretability, and update difficulties—by introducing dual-system design partitioning knowledge into frozen explicit entries and updatable implicit components. Comprehensive experiments demonstrated that ExplicitLM consistently outperforms baseline Transformers across knowledge-intensive tasks, with $20 - 40\%$ improvements in low-data regimes and maintained advantages at scale. Layerwise hit rate analysis confirmed successful memory retrieval directly correlates with prediction accuracy, validating our two-stage differentiable retrieval mechanism. While current implementation requires manual curation of explicit knowledge entries, this limitation points to promising future directions: developing mechanisms to automatically extract and update explicit knowledge from training data while preserving human readability and interpretability. Such advances would enable models to continuously expand verifiable knowledge bases during training, combining statistical learning benefits with transparent, editable knowledge management—crucial for building trustworthy, maintainable AI systems for real-world deployment.

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

## A  APPENDIX

### A.1  REPRODUCIBILITY STATEMENT

We are committed to ensuring the full reproducibility of our work. All experiments presented in this paper can be reproduced using the code and configurations provided in our anonymous repository (`ExplicitLM`). All experiments were conducted on NVIDIA A100 GPUs.

## A.2 AI ASSISTANCE STATEMENT

We declare that AI-based tools were used solely for language polishing purposes in this work. Specifically, after completing the initial draft entirely through human effort, we employed AI assistance exclusively for grammatical refinement and improving the clarity of English expression to meet academic writing standards. The AI tools did not contribute to: (1) the generation or development of research ideas, including the core concept of ExplicitLM and the memory bank mechanism; (2) the design of experiments or methodology; (3) the analysis or interpretation of results; (4) the drafting of original content or scientific arguments; or (5) any mathematical derivations or technical contributions. All intellectual contributions, from conceptualization to initial manuscript preparation, were performed by the human authors. The use of AI was limited to post-writing language enhancement, similar to traditional proofreading services, ensuring that non-native English speakers can present their research with appropriate linguistic quality while maintaining complete authorship and originality of the scientific content.

