# OpenReview forum: "ExplicitLM: Decoupling Knowledge from Parameters via Explicit Memory Banks"
_ICLR.cc/2026/Conference — Submitted to ICLR 2026_

### Official Review · Reviewer_et9n · 2025-10-28

**Soundness:** 2
**Presentation:** 1
**Contribution:** 2
**Rating:** 2
**Confidence:** 4

**Summary:**

This works proposed a learnable memory augmention with explicit (language) and implicit (parametric) knowledge components attached to each layer of a transformer model via a gumbal softmax based retrieval mechanism. The architecture, the transformer and the memory augmentation, is trained jointly on 3 specialized tasks to show the effectiveness of the augmentation. Good improvements are shown against the vanilla transformer baseline, and experiments are designed to analyze the effecacy. Although the design possess novelty to a certain extent, the significance is largely overclaimed. First, it is not a standard language model that generates natural language. Second, the update of the explicit knowledge memory is introduced to be the key to solve the limitations of current "memory frozen" LLMs, which should be core of the proposed ExplicitLM, but is left to future works. Some key architectural designs are missing in the paper. Even if the scale of the implicit memory parameters is in order of O(sqrt(N)), it is still hard to scale in realistic cases.

**Strengths:**

1. The design of the learnable knowledge memory and the potentially updatable (in introduction but not in the current implimentation) explicit is novel and may inspire future works along this line.
2. The improvements in the 3 experiments are significant compared to vanilla transformer.

**Weaknesses:**

1. ExplicitLM is not a standard language model that generates natural language, since the backbone is a transformer trained from scratch via very small datasets.
2. The update mechanism of the explicit knowledge memory is the key to solve the limitations of current "memory frozen" LLMs in as stated in introduction and line 201, which should be core of the proposed ExplicitLM, but is left to future works.
3. The effectiveness is diminishing while the datasets grows bigger. This seems to indicate, given the LLM scale of training data, the proposed method will have little advantage.
4. For the memory hit rate section (4.4), it does not explain an important question: is it just for explicit memory items or also for Implicit memory items? How do you know if it hit the correct implicit memory item?
5. According to Sec4.4 and fig.3, there are roughly 30% cases no correct memory is hit but still the model generates a correct answer. How is that possible. Especially since 30% is not a small percentage, does that mean hit rate is not that important? Table 2 also shows much smaller improvements given the oracle retrieval comparing to the huge boosts in table 1, and it seems also to indicate the hit rate does not have much impact. If the correct retrieval is not that important, then which part of the memory produces the huge boosts in table 1?

3. Missing key model details.
(1) How the retrieved memory embedding is fused with each layer of the transformer (the circle in fig.1)? This is a critical design component, but not explained. Is the fusion process the same for the implicit and explicit memory embeddings? Also not explained.
(2) The detailed parameter settings on the memory augmentation module as well as the backbone transformer are not provided, which are critical to estimate the scalability and cost-efficiency.
(3) The design of the query network is missing.

Minor points:
1. For Eq. (3) and (4), are the two losses defined on each layer so that there actually should be a summation across the layers? Only in this way, they can be added to L_CE in rigorous mathematics.
2. How sensitive the results are against the hyper parameter (the lambdas in the total loss) change is not analyzed.
3. in Fig.2, should "top K" be "top I"?
4. In line 442, "Based on the critical layers identified in Section 4.5", do you mean section 4.4?

**Questions:**

Why not use LLM as the backbone model (frozen or updatable via fine-tuning) and train only the memory module parameters?

---

### Official Review · Reviewer_hDNy · 2025-10-28

**Soundness:** 2
**Presentation:** 3
**Contribution:** 3
**Rating:** 6
**Confidence:** 4

**Summary:**

The paper introduces ExplicitLM, an approach that incorporates explicit and implicit memory banks into the transformer architecture. Specifically, the authors define a memory bank tensor that stores external knowledge but which can retrieved during generation without the necessaity for external knowledge bases. The training process is end-to-end differentiable and experiments show that ExplicitLM significantly improves over the transformer baseline in knowledge-intensive tasks. Further, the authors support their contribution with selected ablation about the hyperparameters introduced in their approach.

**Strengths:**

- The paper is well written and easy to follow. It has a good and detailed technical descripton of the approach that makes it easy to understand what's happening and how ExplicitLM is optimized. The contribution approaches the problem of retrieval augmented generation in a new way by fundamentally rethinking how we can represent external knowledge bases.
- Empirical results show strong improvements over transformer baseline of up to 43.6%. These observations hold in several categories including object detection, relation reasoning and fact verification.
- The ablations are well chosen to help understand chosen hyperparameters and underlying mechanism of ExplicitLM showing trade-off between different ratios of explicit and implicit knowledge.

**Weaknesses:**

- The evaluations are limited to transformer baseline only which makes the contribution less convincing because the transformer architecture is not particularly designed of knowledge retrieval in the first place. The contribution of the paper could be strengthend by comparing ExplicitLM against other baselines such as simple RAG. Other ablations here, such as runtime or scaling performance, may further strengthen the contribution.
- The dataset construction and particularly how the memory bank is setup remains unclear to me. The authors state which datasets are used for it but many important questions such as (1) what data goes into the frozen memory initially, (2) is the memory bank limited to knowledge triplets (as far as I understood it), (3) selection criteria is vaguely formulated or (4) how this dataset is employed for model training, especially for the Memory Relevance Loss that needs query-document pairs for optimization.
- The evaluation paradigm lacks details: Closely related to previous point (4), it remains unclear to me, where the queries for evaluation come from. I am curious why the author haven't chose publically available benchmarks to test their model on comparable datasets.
- The experimental setup section does not include training configuration and important hyperparameters.

**Questions:**

- Citations need to be updated using \citep or similar, currently many citations are implemented using \citet command.
- Variable M needs to be correct in line 156.
- I think a qualitative illustration about what's happening during the retrieval step could help the reader to understand the difference between the implicit and explicit memory stored in the memory bank.

---

### Official Review · Reviewer_nsTD · 2025-10-31

**Soundness:** 1
**Presentation:** 2
**Contribution:** 1
**Rating:** 2
**Confidence:** 4

**Summary:**

This paper attempts to address a meaningful task: knowledge staleness in large language models. However, key details are vague, and comparative experiments are severely lacking. In summary, `the overall completeness of this paper is quite low, and substantial improvements are needed`.

**Strengths:**

1. The research aim of this paper is forward-looking, i.e., building a readable, inspectable, and modifiable explicit memory bank.
2. Explicit division of the memory into different parts is interesting; however, the names for these two types of memory need to be carefully considered.

**Weaknesses:**

1.  **Limited Contribution**
    **Explicit Memory:** Similar concepts exist in Memory3, MemoryLLM, memory modules in model editing (e.g., MEMIT), and earlier Memory Networks.
    *   **Product Key Retrieval:** Seems to directly adopted from works like "Mixture of a Million Experts".
    *   **Readable Memory:** Conceptually similar to retrieving raw text in RAG.

    Consequently, while the paper integrates several existing ideas, `the integration itself is superficial, and the resulting architecture lacks cohesion.`
2.  **Limited Breadth of Evaluation and Baselines:**
    *   **Lack of Critical Comparisons:** Comparing only against a standard Transformer baseline is **severely insufficient**. It is **essential** to compare against state-of-the-art relevant methods, such as:
        *   **Memory related methods** e.g., MemoryLLM, Peripheral Memroy for LLMs etc.
        *   **RAG** or **RETRO:** As alternative paradigms for incorporating external knowledge.
        *   **Model Editing Methods** To evaluate advantages in update precision and locality.
    * **Insufficient Validation of the "Updatability" Promise**: The title and introduction emphasize "Decoupling Knowledge" for easier updates, yet the experiments do not demonstrate any actual knowledge update operations (add, delete, modify) or their effects.
    *   **Narrow Task Focus:** The evaluation tasks (Object Prediction, Relation Reasoning, Fact Verification) primarily test the model's ability to *recall* knowledge from the memory bank. They fail to adequately demonstrate the architecture's capability on more complex, general-purpose tasks like **general language understanding, reasoning, or dialogue**, which are closer to real-world applications.
3.  **Ambiguity Regarding Engineering Overhead and Feasibility:** While efficient retrieval is emphasized, the practical costs of the million-scale memory-**storage overhead, pre-training cost, and inference latency**—are not thoroughly discussed. This raises questions about its practical deployability.
4.  This paper suffers from informal use of in-text citations (author-prominent citation? information-prominent citation?), which detracts from its academic rigor. The symbol of memory in Line 156 is wrong (|M|=N? or |$\mathcal{M}$|=N?).

Due to the **lack of comparisons with the most relevant works** and **shortcomings in the breadth of evaluation and validation of the update mechanism**, the claimed contributions (especially the "Decoupling" for updatability) are not yet fully substantiated in its current form.

**Questions:**

1. What are the most critical distinctions and advantages of ExplicitLM compared to the most relevant works, e.g., MemoryLLM, WISE, Memory3, Peripheral Memroy for LLMs?
2. The definition of memory is confusing (156-157, 160-161). The process of converting these discrete indices into continuous embedding vectors for use within **each transformer layer** is not specified.
3. Does the framework support **precise, rapid updating or deletion of specific knowledge entries**? If so, what is the efficacy and efficiency of such operations?
4. What is the quantitative impact of incorporating this memory bank on **pre-training cost and inference speed**? How much additional storage/time is required for the entire memory bank? It is also difficult for me to find out which LLM you used in this paper?
5. How to initialize the frozen explicit knowledge? How can errors in the initial knowledge be corrected? Does this dependency limit its applicability?

`If there is still time, could you briefly clarify the following question?`
6. Why were other memory-based baselines and RAG not included? Can you demonstrate performance on proper knowledge editing benchmarks (e.g., CounterFact, ZsRE) to prove its general applicability and utility?

## References:
Memory3: Language Modeling with Explicit Memory
MEMORYLLM: Towards Self-Updatable Large Language Models
WISE: Rethinking the Knowledge Memory for Lifelong Model Editing of Large Language Models
Peripheral Memory for LLMs: Integration of Sequential Memory Banks with Adaptive Querying

---

### Official Review · Reviewer_y9Zk · 2025-10-31

**Soundness:** 1
**Presentation:** 3
**Contribution:** 2
**Rating:** 2
**Confidence:** 3

**Summary:**

This paper proposes a new architecture with an explicit, human interpretable large scale memory bank with a differentiable two-stage retrieval mechanism that enables end-to-end training while maintaining discrete knowledge selection. The external memory bank is divided into two partitions-  frozen explicit facts (20%) and learnable implicit patterns (80%). The hierarchical 2 stage retrieval approach with stage 2 using gumbel softmax ensures end to end differentiability. This enables  joint optimization of retrieval and generation through differentiable selection. They have combined language modeling loss with retrieved memory relevance loss and memory diversity loss. The paper also proposes solution to gradient discontinuity issue by adopting EMA from VQ-VAE, which allows smooth transition for knowledge update.
Main argument
The paper does a good job setting up the problem. Knowledge augmentation/ update/ editing is an open research problem. The proposed architecture aimed to address this. However, the experiment set up is not rigorous enough to demonstrate that this approach really works. There is almost no details on what kind of model is used as baseline, other than saying it’s a standard transformer model. Second, while it does a goos job in setting up the evaluation tasks, there is no discussion about what has been used as an evaluation set. No result on standard benchmarks are presented. While the paper presents results of perfect retrieval, they also did not compare their results with RAG or other related work such as “Memory Layer at scale”.  It is not a fair comparison with vanilla transformer, especially when this approach augments the model with explicit knowledge initialized(and kept frozen) with curated factual information. The paper also doesn’t say what % of additional parameters does it add compare to the baseline.
It would also be good to see
There are so many missing details it is difficult to draw many conclusions:
1. More details about baseline is needed
2. The value add of the proposed architecture is not obvious without comparing with RAG baselines, especially when this architecture is augmented with external knowledge
3. More details about the evaluation tasks(with examples) are needed. The object prediction tasks seemed less challenging
4. More details about evaluation dataset is needed, specially making sure there is no data leak, wiki has many duplicates
5. The paper didn’t present results in any of the standard benchmarks that re used for memory or continual learning, such as SImpleQA, NaturalQA
6. Many related works are not acknowledged or compared to. Few related works are - Memory Layers at Scale, Continual Learning via Sparse Memory Finetuning, Learning facts at scale with active reading
7. How much more trainable parameters are being used here?
The results are not convincing given the above details are missing. My recommendation is reject.

Things to improve the paper that did not impact the score:
1. As the paper seems to aim solve knowledge update with implicit memory, it would be good to see results on such tasks. Please see the Active reading paper mentioned above.
2. In the “perfect data retrieval” section, no results are provided for “low data regime”, which is one of the key area the architecture is aiming to solve

**Strengths:**

The paper does a good job setting up the problem. Knowledge augmentation/ update/ editing is an open research problem.

**Weaknesses:**

The experiment set up is not rigorous enough to demonstrate that this approach really works. There is almost no details on what kind of model is used as baseline, other than saying it’s a standard transformer model. Second, while it does a goos job in setting up the evaluation tasks, there is no discussion about what has been used as an evaluation set. No result on standard benchmarks are presented. While the paper presents results of perfect retrieval, they also did not compare their results with RAG or other related work such as “Memory Layer at scale”.  It is not a fair comparison with vanilla transformer, especially when this approach augments the model with explicit knowledge initialized(and kept frozen) with curated factual information. The paper also doesn’t say what % of additional parameters does it add compare to the baseline.

**Questions:**

1. More details about baseline is needed
2. The value add of the proposed architecture is not obvious without comparing with RAG baselines, especially when this architecture is augmented with external knowledge
3. More details about the evaluation tasks(with examples) are needed. The object prediction tasks seemed less challenging
4. More details about evaluation dataset is needed, specially making sure there is no data leak, wiki has many duplicates
5. The paper didn’t present results in any of the standard benchmarks that re used for memory or continual learning, such as SImpleQA, NaturalQA
6. Many related works are not acknowledged or compared to. Few related works are - Memory Layers at Scale, Continual Learning via Sparse Memory Finetuning, Learning facts at scale with active reading
7. How much more trainable parameters are being used here?

---

### Meta-Review · Area_Chair_upeG · 2026-01-07

**Summary:**

The paper proposes ExplicitLM, a transformer with a large external memory bank and a differentiable retrieval mechanism to select discrete knowledge while training end-to-end. Reviewers agree the problem (knowledge staleness and editable memory) is important and the idea is interesting. However, the evidence is not strong enough to support the claimed benefits.

Pros
1. Addresses an important problem: keeping model knowledge up to date
2. Clear high-level motivation and an interpretable memory design (explicit and implicit parts)
3. Shows improvements over a vanilla transformer baseline on the paper’s tasks

Cons
1. Too few and too weak baselines: mainly compared to a standard transformer, missing comparisons to RAG/RETRO, prior memory methods, and model editing approaches
2. Updatability is a key claim but not demonstrated: no clear experiments on adding/deleting/modifying knowledge or standard editing benchmarks
3. Missing experimental details: unclear baseline setup, memory construction, evaluation data/queries, and data leakage controls
4. Practical cost is unclear: no solid reporting of extra parameters, storage, training cost, or inference latency
5. Some key implementation details are missing (how memory is fused into the transformer, query network), making reproduction difficult

**Reviewer Scores:**

N/A

---

### Decision · Program_Chairs · 2026-01-26

Reject